# The reaction-diffusion basis of animated patterns in eukaryotic flagella

**James F. Cass** [1] **& Hermes Bloomfield-Gadêlha** [1] ✉

The flagellar beat of bull spermatozoa and *C. Reinhardtii* are modelled by a minimal, geometrically exact, reaction-diffusion system. Spatio-temporal animated patterns describe flagellar waves, analogous to chemical-patterns from classical reaction-diffusion systems, with sliding-controlled molecular motor reaction-kinetics. The reaction-diffusion system is derived from first principles as a consequence of the high-internal dissipation by the flagellum relative to the external hydrodynamic dissipation. Quantitative comparison with non-linear, large-amplitude simulations shows that animated reaction-diffusion patterns account for the experimental beating of both bull sperm and *C. Reinhardtii*. Our results suggest that a unified mechanism may exist for motors controlled by sliding, without requiring curvature-sensing, and uninfluenced by hydrodynamics. High-internal dissipation instigates autonomous travelling waves independently of the external fluid, enabling progressive swimming, otherwise not possible, in low viscosity environments, potentially critical for external fertilizers and aquatic microorganisms. The reaction-diffusion system may prove a powerful tool for studying pattern formation of movement on animated structures.

As a paradigm for studying pattern-formation in nature, Reaction-Diffusion (RD) models have proliferated across the sciences since Turing's seminal paper[1]. Motivated by animal markings, he described a diffusion-driven instability of a spatially homogeneous state of two morphogens, that generates heterogeneous equlibria, such as spots or stripe patterns (Fig. 1a). *Non-equilibrium* phenomena, such as stable oscillations, also feature in many RD models with a persistent source of energy; for example, the Belousov-Zabotinsky (BZ) reaction[2], or predator-prey systems[3], among other non-morphogenetic reaction systems[4,5]. Oscillations can persist if a smaller part of the system is isolated without diffusion. Coupling the isolated parts, via diffusion, can entrain oscillators with non-trivial phase differences. This drives, for example, intricate spiral waving patterns in the BZ reactions[6]. However, despite the universality of RD systems whilst describing a bewildering array of patterns across science, it is still unclear whether RD theory can be expanded to animated non-equilibrium patterns in nature— spatio-temporal patterning of self-organised, shape-shifting structures; the archetype of which is the spontaneous beating of eukaryotic cilia and flagella, depicted in Fig. 1b[7].

Eukaryotic cilia and flagella are slender cellular appendages that spontaneously generate propagating waves of flagellar curvature[8] (Fig. 1b). This time-irreversible patterning of the flagellum is crucial to evade Purcell's scallop theorem and enable microswimmers, such as mammalian spermatozoa to swim, or cilia in the respiratory tract to generate flows that pump mucus[9–11]. The internal core structure of the flagellum, the axoneme, consists of a central pair of microtubule doublets with an outer cylindrical structure of nine doublets[12], known as the 9 + 2 structure (see Fig. 2a). Dynein motor proteins, regularly anchored along the doublets, exert force couples by cross-linking between neighbouring filament pairs, forcing the sliding of doublets relative to each other—forming the basis of the so-called sliding control hypothesis[7], as discussed below. Bending of the flagellum occurs when activity of the dyneins is coupled with sliding constraints[7], converting relative sliding to bending. Stable periodic wave propagation,

[1]School of Engineering Mathematics and Technology, and Bristol Robotics Laboratory, University of Bristol, Bristol, UK.
✉e-mail: hermes.gadelha@bristol.ac.uk

**a** spatial patterns

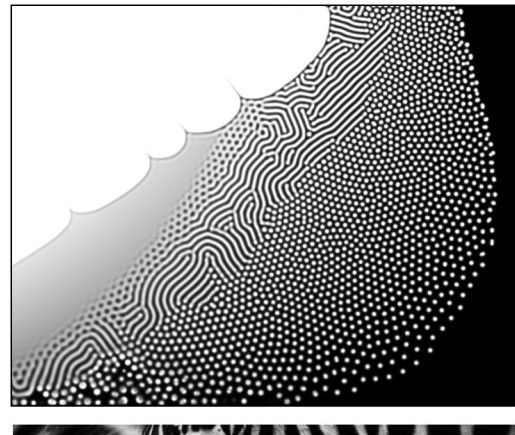

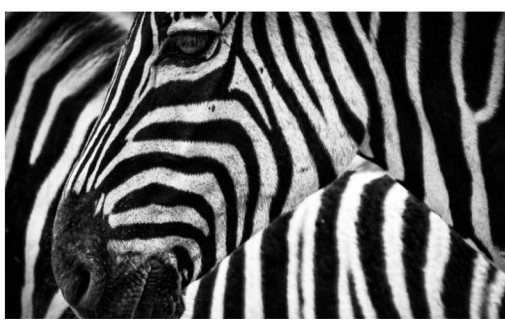

**b** spatio-temporal patterns

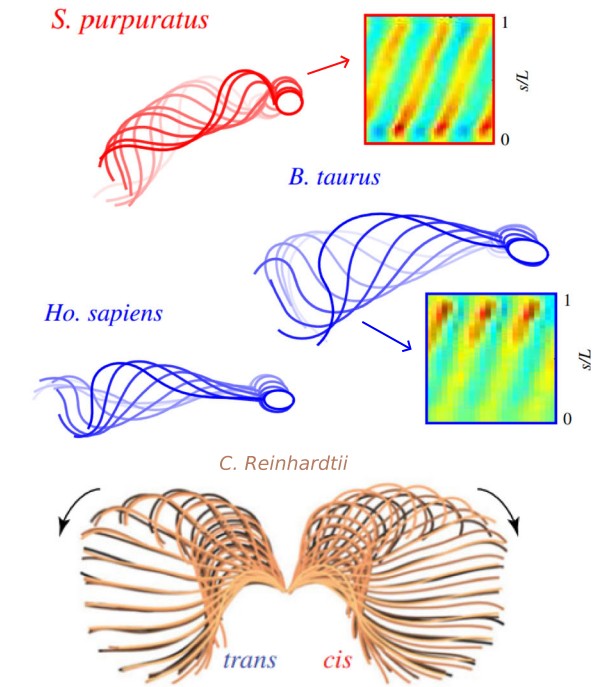

**Fig. 1 | Pattern formation in reaction-diffusion systems. a** Generic reaction-diffusion patterns generated in a web browser based simulator (apps.amanda-ghassaei.com/gpu-io/examples/reaction-diffusion). The striped central region resembles the stripes of the zebra (image obtained from Pixabay, 2023). Such animal markings motivated Turing when deriving his model system[1]. **b** Animated spatio-temporal patterns in eukaryotic flagella. The beating patterns of human, bovine and sea urchin (S. purpuratus) spermatozoa are shown (figure adapted with permission[47], Copyright 2020 Royal Society, 1371128-1) along with the breaststroke of the two flagella (cis and trans) of the green alga *Chlamydomonas reinhardtii* (figure adapted from ref.[78], CC BY 3.0). Kymographs show spatio-temporal stripe patterns of flagellar curvature that we show can be modelled with reaction-diffusion dynamics.

however, requires collective activity of the dyneins and the underlying mechanism of self-organisation remains an open area of research.

In 1975, Brokaw suggested in his seminal work that coupled "local shear oscillators", i.e., small sections of a flagellum that undergo oscillatory shear deformation, might be sufficient to explain global flagellar bending waves[13], if "spatial phase differences could be self-organised" that would induce propagation[14]. Whether such a mechanism could exist has remained an open question [*ibid*]. Here, we show that animated beating patterns in eukaryotic flagella (Fig. 1b) can be understood physically in the manner Brokaw envisioned, and are congruent with patterns displaying phase differences from classical oscillatory reaction-diffusion systems, which typically model biochemical patterning[3], rather than animated mechanical shaping[15]. In contrast to the interpretation of chemical species freely reacting and diffusing in space, our model system describes the reaction-kinetics of molecular motors that are anchored in place on the filaments, but the shear deformation that they generate from sliding deformations can "diffuse" away via the bending elasticity of the axoneme that couples neighbouring elements (Fig. 2).

Since fitting complex mathematical models with noisy spatio-temporal data is challenging, the linear regime of small amplitudes has been exploited until now to obtain flagellar waveforms that can be matched via analytical solutions in three seminal papers[16–18]. In this approach, the chemical reactions that power the dynein kinetics are approximated by a geometrical response coefficient, rather than explicitly modelled, leaving just a single linear "sperm equation" to be solved[16]. The linear approximation was crucial to show how self-organised flagellar bending waves could arise via a dynamic instability, and the influence of boundary conditions on the resulting beating patterns[19,20]. Our novel contribution is to show that beyond the linear

regime, where realistic large amplitude flagellar beat patterns are attained, the dominant balance of moments shifts towards oscillatory reaction-diffusion dynamics, with wide-ranging consequences, suggesting that extrapolation of conclusions based on scaled-up linear solutions (since the wave amplitude is not determined) can be problematic. We derive from first principles the reaction-diffusion theory and show quantitatively that the resulting animated patterns can accurately mimic the large amplitude swimming gaits of two eukaryotic microorganisms (See Supplementary Videos 1 and 2), in contrast to previous linearised models[17].

The reaction-kinetics of dynein molecular motors and their response to external forces is the focus of intense debate[14]. Mechanical feedback is a possible regulatory mechanism of dynein attachment and detachment that localises activity to either side of the axoneme (Fig. 2)[21], driving shear and consequently bending. Static images from cryo-electron tomography show dynein in different conformations in different regions of the sea urchin sperm axoneme, consistent with the switching hypothesis[21], while not definitive. Feedback has been hypothesised to come from local curvature[22–24], relative shearing/sliding[16,20,25–27] or transverse forces to the doublets[28–30] (see refs. 14,31 for reviews). A separate line of enquiry considers the steady action of dynein that generates cilia-like oscillations via a flutter instability[32,33] without requiring dynamical switching of dyneins. Obtaining solutions for these systems is challenging due to the need of balancing non-local moments arising from fluid-structure interactions with the molecular-motor activity[26,27]; referred to here as *chemo-ElastoHydrodynamic* (chemoEH) flagellar models. This difficulty, along with a high-dimensional parameter space, may be why the quantity of modelling papers far outweighs the number of quantitative comparisons of waveform predictions with experiments−to date, only two studies

have directly contrasted the sliding control hypothesis against waveform data[16,17].

The specific molecular motor control mechanism we will employ[26], known as load-dependent detachment, belongs to the *sliding-controlled* class of models[16,19,20]. The detachment rate of motors is modelled as depending exponentially on the load that they feel in the direction of sliding motion, which has been observed experimentally for kinesin[34]. With two teams of antagonistic motors engaged in a tug-of-war (Fig. 2)[25,35], a positive feedback loop is possible whereby if one team gains the upper hand, the load per motor decreases allowing more motors to attach. For the losing team, the load increases leading to more detachment. In an influential paper[16], a linearised model of sliding-control (the single equation already described) was shown to fit nonlinear beating patterns of bull sperm with remarkable accuracy. Later, however, it was shown that the sliding-controlled model could not account for the beating of *C. reinhardtii*[17]. In contrast, a dynamic curvature control response (i.e., proportional to the time derivative of curvature) was able to fit the data [*ibid*]. Current evidence suggests, then, that there may be different mechanisms by which the beat is generated in the approximately 50 μm flagella of bull sperm compared to the 10 μm cilia of *C. reinhardtii*[17]. On the other hand, while the load-dependent detachment mechanism gives sliding control a plausible molecular basis, no such mechanism of curvature sensing by dynein has been proposed, nor observed to date[17,18,36].

Until now, experimental fitting has not been attempted with solutions of a model system with specific molecular cross-bridge kinetics, and not restricted to the small-amplitude linear regime[16–18], an important test of the validity of these earlier results. To do so requires comparison with numerical simulations, since analytical solutions are generally not available for large amplitude motion. Oriola et al.[26] considered the limit of small curvature to recast the full chemoEH system into a geometrically linear, *reaction-hyperdiffusion* system - where "hyper" here is linked with the fact that fourth-order space derivatives appear instead of second-order for diffusion problems[37] (A further example of a reaction-hyperdiffusion system is given in ref. 30, in the context of the "geometric clutch"). Ref. 26 proposed a tug-of-war dynein kinetics of the flagellum that led to spontaneous and nonlinear saturation of unstable modes in simulations without the need of ad-hoc nonlinearities. This prediction was subsequently confirmed via further numerical simulations of the full chemoEH system[27]. We go further than these studies by exploring simulations of a freely-swimming spermatozoon, and we derive the simpler RD model which makes comparing simulations with experiments tractable.

Mathematical models of flagellar beating have tended to derive the governing equations using a sliding-filament mechanism that considers the equal balance of active, elastic, and viscous (hydrodynamic) moments along the flagellum[22,24,38]. Murase, in contrast, studied an excitable dynein model assuming zero external viscosity, that propagated waves along the flagellum when periodically forced at the base[39,40]. Brokaw found similarly that the wavelength of simulated flagellar oscillations could be stabilised with enough internal dissipation relative to external viscosity[23,41]. The physical implication of a high internal/external viscosity ratio can be understood by considering the different approximations that are commonly used for the hydrodynamic force on the flagellum, for example: resistive force theory (RFT)[42] (used in refs. 26,30,43) considers only the force contribution from a locally straight section around each point of arclength, whereas slender-body theory (used in ref. 27) includes interactions with more distant parts of the flagellum. In either case, intuitively, hydrodynamic influence is felt from beyond the directly neighbouring shearable elements. This is distinct from the nearest-neighbour coupling derived from bending elasticity that remains upon neglecting external viscosity.

The no external viscosity assumption was used previously only for ease of numerical computation[39,40]. Recent experimental studies, however, provide physical, rather than computational, reasons to consider this approximation in greater detail, suggesting that the contribution of external viscous moments to the overall balance may indeed be small[44,45]. This is counter-intuitive; after all the flagellum must do appreciable work on the fluid in order to swim, and the energy generated by the motors must be dissipated somewhere to sustain stable oscillations. Accordingly, internal dissipation of energy has been calculated to be significantly larger than external dissipation in experiments with beating tethered mouse sperm[45], and *Chlamydomonas reinhardtii*[44]. Similarly, it was recently argued that external viscous forces must be small compared to internal elastic forces, also for *C. reinhardtii*, in a wide range of biochemical conditions[18]. There could be varied sources of internal dissipation, and previous simulation studies have included such terms in the governing equations[23,24,30]. Nandagiri et al.[45] directly measured, however, that a significant proportion of the energy generated by the motors is dissipated within the workings of the molecular motors themselves. We capitalise on these findings to derive the RD model as the small external, and high internal, dissipation limit observed empirically for the axoneme[44,45]. We show how dissipation of energy over the beating cycle only occurs within the molecular motors as they cycle through conformational changes[25], and other potential sources of dissipation from the passive axoneme, such as microtubule bending friction, may be neglected[23,30,39,44]. Little is known when internal dissipation dominates self-organised beating at the geometrical nonlinear level, in which case we demonstrate that the system may be described with chemical reactions and elastic interactions only.

In this paper, we derive a geometrically nonlinear reaction-diffusion system for animated patterning in eukaryotic flagella that is valid far from the quiescent equilibrium state. We show that the chemo-elastic RD system is a natural consequence of the high internal dissipation limit of the full chemo-elastohydrodynamic (chemoEH) system, which embodies the fluid-structure interaction of a freely swimming microswimmer consisting of a cell body with attached flagellum, driven by tug-of-war reaction-kinetics. As such, the RD theory can be derived without invoking linearisations of the nonlinear dynamics and/or its geometry. Self-organised propagating waves are a feature of the RD system. We demonstrate with numerical simulations (Python code provided[46]) the equivalence of the beating generated by the reaction-diffusion system and the beat of the free-swimmer, in the region where the RD approximation is valid. For the first time, we fit a fully nonlinear, large-amplitude and self-organised cross-bridge kinetics model of the flagellar beat with experimental data from the literature. Thereafter we are able to make a comparison beyond the emergence of the flagellar beat, of the self-organised swimming trajectory. This leads to conclusions divergent from small amplitude theory[16,17,20]. Most notably, the RD theory is capable of reproducing accurately the characteristic beating patterns of evolutionarily distant eukaryotic species: *Chlamydomonas reinhardtii* (wild-type and mbo2-mutant)[17] and bull spermatozoa[47]. These differ markedly in the length of their flagella/cilia, their axonemal structure, flagellar ultrastructure, cell morphology, and function.

## Results

In the Supplementary Information (SI), we derive the model system of equations for a free-swimming spermatozoon, referred to as the chemo-ElastoHydrodynamic (chemoEH) flagellar model. From this system, we derive our flagellar Reaction-Diffusion (RD) model as the limit in which viscous moments are small compared with passive shearing moments. In this section and the following we aim for physical understanding of the RD model and its solutions.

### Reaction-diffusion model of the flagellar beat

In order to model two-dimensional beating of a flagellum, we first project the three-dimensional axoneme onto the plane of bending; see

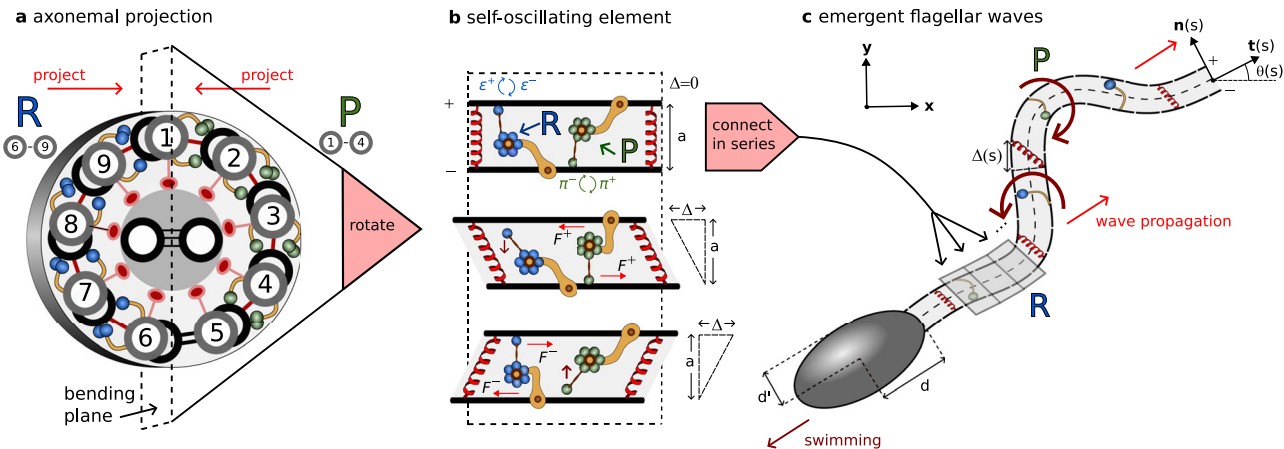

**Fig. 2 | Modelling overview. a** Cross section of an element of the flagellar axoneme, with 9 + 2 structure of microtubules cross-linked by passive nexin and radial spokes (red) and active dynein molecular motors (blue/green generating principal (P)/ reverse (B) bends). **b** Two-dimensional projection of the axoneme into the plane of bending onto two filaments (+/−) at fixed separation $a$. Distribution of dynein represented as single motors attaching/detaching with rates $\pi^{\pm}/\epsilon^{\pm}$, generating oppositely directed shear displacement $\Delta$ via forces $F^{\pm}$. **c** When coupled in series via bending elasticity local oscillations of shearable elements induce emergence of flagellar waves characterised by tangent angle $\theta(s)$ in arclength $s$.

Fig. 2a. Each small element of the model flagellum (Fig. 2b) consists of two filaments which are separated at a fixed distance $a$ and connected by active and passive cross-linking elements[20]. When bound, active molecular motors on either side exert oppositely directed tangential forces $F^+$ and $F^-$, which cause the element to deform in shear, quantified by the sliding displacement $\Delta$. Blue/green motors, corresponding to forces $F^-/F^+$, are responsible for positive/negative shear deformations (since dynein motors walk towards the basal end of the flagellum, assumed left in Fig. 2b). The load per motor is modelled by a linear force-velocity relationship $F^{\pm} = f_0(1 \pm \Delta_t/v_0)$, where $f_0$ is the force at stall ($\Delta_t = 0$) and $v_0$ is the velocity when the motors produce no force. The dyneins are thus force generators with internal damping elements associated with a constant drag coefficient $\xi^m = f_0/v_0$[25].

Rather than a single pair of motors per element as depicted in Fig. 2b, we consider a constant density $\rho$ and the proportions of motors on the plus and minus filaments $n^+$ and $n^-$ which are in the bound state. The tangential motor force per unit length exerted on the plus filament is then a tug-of-war given by $f^m = \rho(n^+ F^- - n^- F^+)$. Nexin links resist shearing elastically and are modelled by a force per unit length $f^s = -K\Delta$ where $K$ is a Hookean spring constant. The total tangential force per unit length $f^t = f^m + f^s$ on the plus filament from cross-linking elements is

$$f^t = \rho f_0 \left[ \tilde{n} - \bar{n}\frac{\Delta_t}{v_0} \right] - K\Delta. \tag{1}$$

where $\tilde{n} = n^- - n^+$ and $\bar{n} = n^- + n^+$. The dynamics of the bound motor populations $n^+$ and $n^-$ are given as in ref. 26 by the two-state system $n_t^{\pm} = \pi^{\pm} - \epsilon^{\pm}$, where $\pi^{\pm}$ and $\epsilon^{\pm}$ represent attachment and detachment rates, respectively (Fig. 2b). More specifically, the motor populations evolve according to the rate equations

$$n_t^{\pm} = \pi_0(1 - n^{\pm}) - \epsilon_0 n^{\pm} \exp\left[ \frac{f_0}{f_c}\left(1 \pm \frac{\Delta_t}{v_0}\right) \right]. \tag{2}$$

While attachment $\pi^{\pm} = \pi_0(1 - n^{\pm})$ is simply related with the proportion of unbound motors, the detachment rate $\epsilon^{\pm} = \epsilon^{\pm}(\Delta_t)$ implements a sliding-controlled feedback mechanism; the base rate $\epsilon_0$ is modulated exponentially by the ratio of the motor force $F_{\pm}$ to a characteristic unbinding force $f_c$.

The elements are connected in series, as in Fig. 2c, such that $\Delta(s)$ and $n^{\pm}(s)$ become functions of arclength $s \in [0, L]$ (considering the

continuous limit of small elements). Dynein activity leads to differential tension in the two filaments, which generates an active moment about the centerline of the structure. The flagellum bends in response, its orientation at each point characterised by the angle $\theta(s)$ between the tangent vector $\mathbf{t}(s)$ to the centerline and the $x$-axis of the laboratory fixed frame of reference. Figure 2 uses the common terminology of the principal (P) bend generated by negative-shearing green motors, and the reverse (R) bend generated by positive-shearing blue motors[48]. If the spacing $a$ is constrained to remain constant, the sliding displacement can be shown to be given by $\Delta(s) = \Delta_0 + a\gamma(s)$[20] where $\Delta_0$ accounts for sliding at the basal end and $\gamma(s) = \theta(s) - \theta(0)$ is the shear angle. We only consider $\Delta_0 = 0$ in this paper. The centerline position vector $\mathbf{r}^H(s)$ in the fixed head frame of reference can be constructed from the shear angle via $\mathbf{r}^H(s) = \int_0^s \mathbf{t}\, ds'$ where $\mathbf{t} = (\cos\gamma, \sin\gamma)$.

In our limit of interest, the active moment at a point $M^a = -a\int_s^L f^t ds'$ is only resisted by the elastic bending moment $M^b = B\gamma_s$, where $B$ is the bending rigidity of the flagellar bundle. Any viscous contribution is considered to be small (see Section S2 in SI), so that $M^a + M^b = 0$. Differentiation leads to

$$\left(\frac{a^2\rho f_0}{v_0}\right)\bar{n}\gamma_t = B\gamma_{ss} - a^2 K\gamma + a\rho f_0 \tilde{n}. \tag{3}$$

Note that Eq. (3) does not depend on the lab-frame tangent angle $\theta$ and therefore solutions will describe the shaping of the flagellum relative to the body frame of reference. Finally, we must specify two boundary conditions for the shear angle. We consider (i) no shearing at the base, or $\gamma(0) = 0$ and (ii) no external moments at the distal end, so that $B\gamma_s(L) = 0$. In this framework, the structure at the base of the axoneme exerts a moment that resists any basal shearing, but the body-flagellum coupling only affects the global translation/rotation of the swimmer. If the molecular motors are not present, the moment balance leads to the filament-bundle elastica equation, $B\gamma_{ss} - a^2 K\gamma = 0$, which also captures static configurations of the flagellar counterbend phenomenon[49–51], a reversal of curvature when a passive flagellum is bent with a microprobe[52].

The RD model consists of Eqs. (2) and (3) with the boundary conditions, combining reaction-kinetics with an elastic axoneme (chemo-elastic). For simplicity, we rewrite the governing equations in a

non-dimensional form (see SI),

$$\mu_a \zeta \bar{n} \gamma_t = \gamma_{ss} - \mu\gamma + \mu_a \tilde{n}, \tag{4}$$

$$n_t^{\pm} = \eta(1 - n^{\pm}) - (1 - \eta)n^{\pm}\exp[f^*(1 \pm \zeta\gamma_t)]. \tag{5}$$

Eqs. (4–5) are a system of partial differential equations in the dynamical variables $\gamma, n^+$ and $n^-$, with the form of a generalised *reaction-diffusion* system. This is more clearly seen by writing the above system in the matrix form

$$\mathbf{M}(\mathbf{u})\mathbf{u}_t = \mathbf{D}\mathbf{u}_{ss} + \mathbf{L}\mathbf{u} + \mathbf{N}(\mathbf{u},\mathbf{u}_t) \tag{6}$$

for the state vector $\mathbf{u} = (\gamma, n^+, n^-)$. The entries of the matrices are stated explicitly in Sec. S2 of SI. Here, $\mathbf{M}(\mathbf{u})$ is a state dependent mass matrix, $\mathbf{D} = \mathrm{diag}(1, 0, 0)$ is a diagonal matrix of diffusion coefficients, and $\mathbf{L}$ and $\mathbf{N}$ are linear and nonlinear operators. While canonical RD systems describe reaction and diffusion of all interacting elements, here, instead, the diffusion coefficients of the motor populations $n^{\pm}$ are zero, since they are anchored in place to their respective filaments, Eq. (5). The shear $\gamma$ may diffuse, however, owing to the viscoelasticity of the axoneme that comes from the passive elastic structure combined with molecular motor dissipation, Eq. (4). There may also be experimental situations modelled using this framework where diffusion of molecular motors is present (see discussion). If $\mathbf{D} = 0$, Eqs. (4), (5) only depend on quantities at the point $s$ in arclength. As we will see below, the RD model exhibits temporal oscillations of shear, locally, which interact diffusively to form spatio-temporally animated wave patterns.

## Spontaneous oscillations and diffusion of flagellar shear

We first examine an isolated, freely shearing element of the flagellum (Fig. 2b) that experiences no contact forces and moments from neighbouring material. Since internal forces must cancel, the force balance in the tangential direction on either filament is $f^t = 0$, where $f^t$ is given in Eq. (1) with $\Delta$ and $n^{\pm}$ now treated as scalar quantities. This equation suggests the possibility of spontaneous oscillations−balancing on average the input of energy $\rho f_0 \tilde{n}$ with dissipation in the attached motors $\rho f_0 \bar{n}\Delta_t/v_0$−and we find that oscillatory solutions indeed exist.

The system of ordinary differential equations consisting of $f^t = 0$ and Eq. (2) has an equilibrium solution given by $(\Delta, n^{\pm}) = (0, n_0)$. The value $n_0 = \pi_0/(\pi_0 + \epsilon_0 e^{f^*})$ gives the proportion of motors in the attached state at equilibrium. Hence $\bar{\tau} = (\pi_0 + \epsilon_0 e^{f^*})^{-1}$ is a characteristic time scale of the motor activity. Solving the linearised system about this equilibrium (see Section S3 in SI) reveals a Hopf bifurcation of the non-dimensional ratio $\nu_a = (\rho f_0)/(aK)$ of a characteristic motor force to the resistance to shear. At the critical value, given by $\nu_a^{\mathrm{crit}} = (2\bar{\zeta}n_0\omega_0^2)^{-1}$, the equilibrium solution becomes unstable to small perturbations and the shearing element starts to undergo spontaneous oscillations. Here $\bar{\zeta} = a/(v_0\bar{\tau})$ and the critical frequency of oscillations is given by $\omega_0^2 = (1 - n_0)f^* - 1$. Previous studies have instead used $\tau = (\pi_0 + \epsilon_0)^{-1}$ as the characteristic time scale and $\zeta = a/(v_0\tau)$[26,27,53]. We will make use of both the barred and unbarred quantities.

Figure 3a shows an example of spontaneous oscillations of an isolated element, where an initial condition close to the equilibrium grows and after a transient saturates into limit-cycle oscillations of (i) $\gamma = \Delta/a$ and (ii) $n^{\pm}$. Blue/green motor activity drives positive/negative shear after a delay of approximately one-eighth of a period. Panel (iii) shows the force balance; at the dotted line marked 1 in Fig. 3a, the element is depicted at its maximum negative shear, where the elastic restoring force and the active force exactly cancel so that there is no sliding velocity. From Eq. (2) with $\gamma_t = 0$ we see that the motors will move towards the equilibrium $n_0$, weakening the active force that depends on $\tilde{n}$. At this point, the restoring force will gain the upper

hand and drive the sliding velocity back in the opposite direction. Dotted line 2 shows the situation at a $180°$ phase offset of the oscillation period (Fig. 3a).

Turning back to a flagellum of length $L$, we consider now the case of no sliding feedback mechanism i.e., the detachment rate $\epsilon^{\pm} = \epsilon_0$ is constant. In this special case, the motor populations relax to the equilibrium value $n_0$ and Eq. (3) becomes a diffusion equation (also the heat equation) with a restoring force, $\gamma_t = D\gamma_{ss} - E\gamma$ where $D = (Bv_0)/(2n_0a^2\rho f_0)$ is the diffusion coefficient and $E = (Kv_0)/(2n_0\rho f_0)$ is an effective spring constant. This equation is solved using standard methods with general solution given by $\gamma(s,t) = \sum_k c_k \exp\{-(E + D(k + \frac{1}{2})^2\pi^2/L^2)t\}\sin\{(k + \frac{1}{2})\pi s/L\}$ for constants $c_k$ determined by the configuration when $t = 0$; i.e., the shear distribution "diffuses" exponentially on a *diffusive* timescale that scales with $L^2$, that is shifted when $E \neq 0$. Note this is in contrast with hyperdiffusive relaxations of filaments and filament-bundles elastohydrodynamics that are dominated by external fluid viscosity, where the relaxation time has a stronger length dependence of $L^4$[50,54,55]. Figure 3b shows an example of the diffusive behaviour of shear, where the colouring along the flagellum marks the amount of shear $\gamma$. The filament relaxes from an initially curved configuration, where the elastic energy stored in the bent flagellum is dissipated by the molecular motor crosslinking, as they are driven by elastic restoring forces.

## Emergent flagellar waves

Using linear stability analysis, it can be shown (Section S3 in SI) that the RD model (6) exhibits a Hopf bifurcation. We express our results in terms of the non-dimensional ratio $\mu_a = (a\rho f_0 L^2)/B$ of the motor force $a\rho f_0$ to the elastic bending force $B/L^2$, and the ratio $\mu = (a^2 KL^2)/B$ of the shear resistance to bending resistance (note that $\nu_a = \mu_a/\mu$). The Hopf bifurcation point is $\mu_a^{\mathrm{crit}} = (\pi^2 + 4\mu)/(8n_0\bar{\zeta}\omega_0^2)$ where $\omega_0$ is unchanged; setting the first summand of $\mu_a^{\mathrm{crit}}$ to zero reduces it to the single element case seen above. We measure activity relative to the bifurcation with the parameter $\epsilon = (\mu_a - \mu_a^{\mathrm{crit}})/\mu_a^{\mathrm{crit}}$, as in ref. 26. Solutions for the flagellar shape of the linearised problem take the form $\gamma(s,t) = A\exp[(\alpha + i\omega)t/\tau]\sin(\pi s/(2L))$, growing oscillations multiplied by a spatial sinusoid ($A$ is an undetermined constant). This is a self-organised standing wave where shear oscillations are synchronised in-phase throughout the flagellum. The growth rate $\alpha$ and frequency $\omega$ can be expressed as functions of $\epsilon$:

$$\alpha = \frac{\epsilon\omega_0^2}{2(1 + \epsilon)}, \tag{7}$$

$$\omega^2 = \left[\frac{1 + \epsilon - \frac{1}{4}\omega_0^2\epsilon^2}{(1 + \epsilon)^2}\right]\omega_0^2. \tag{8}$$

We compare the predictions of Eqs. (7) and (8) (solid lines) with simulations of the RD system (6) (filled circles) in Fig. 3c, using parameter values as in refs. 26,27. The predictions follow closely the simulated values for $\epsilon < 0.4$ and the theoretical frequency curve remains a good prediction of the simulated values even far from the bifurcation where the linear theory breaks down.

Simulations of the RD model for 3 values of $\epsilon$ are shown in the columns (i)–(iii) of Fig. 3d. Near to the bifurcation, we observe (approximately) a standing wave as predicted above, but as $\epsilon$ is increased further, we find a change in the space-time animated patterning: bend initiation at the base leading to base-to-tip propagation of flagellar waves with increasing wavespeed (slope of the kymograph in Fig. 3d). Wave propagation corresponds to a relative phase offset in the shear oscillations of neighbouring elements. Simulations of a freely swimming sperm (chemoEH model) are shown in the final two rows of Fig. 3d. The commonly used sperm number $\mathrm{Sp} = L/l^h$ is the ratio of the

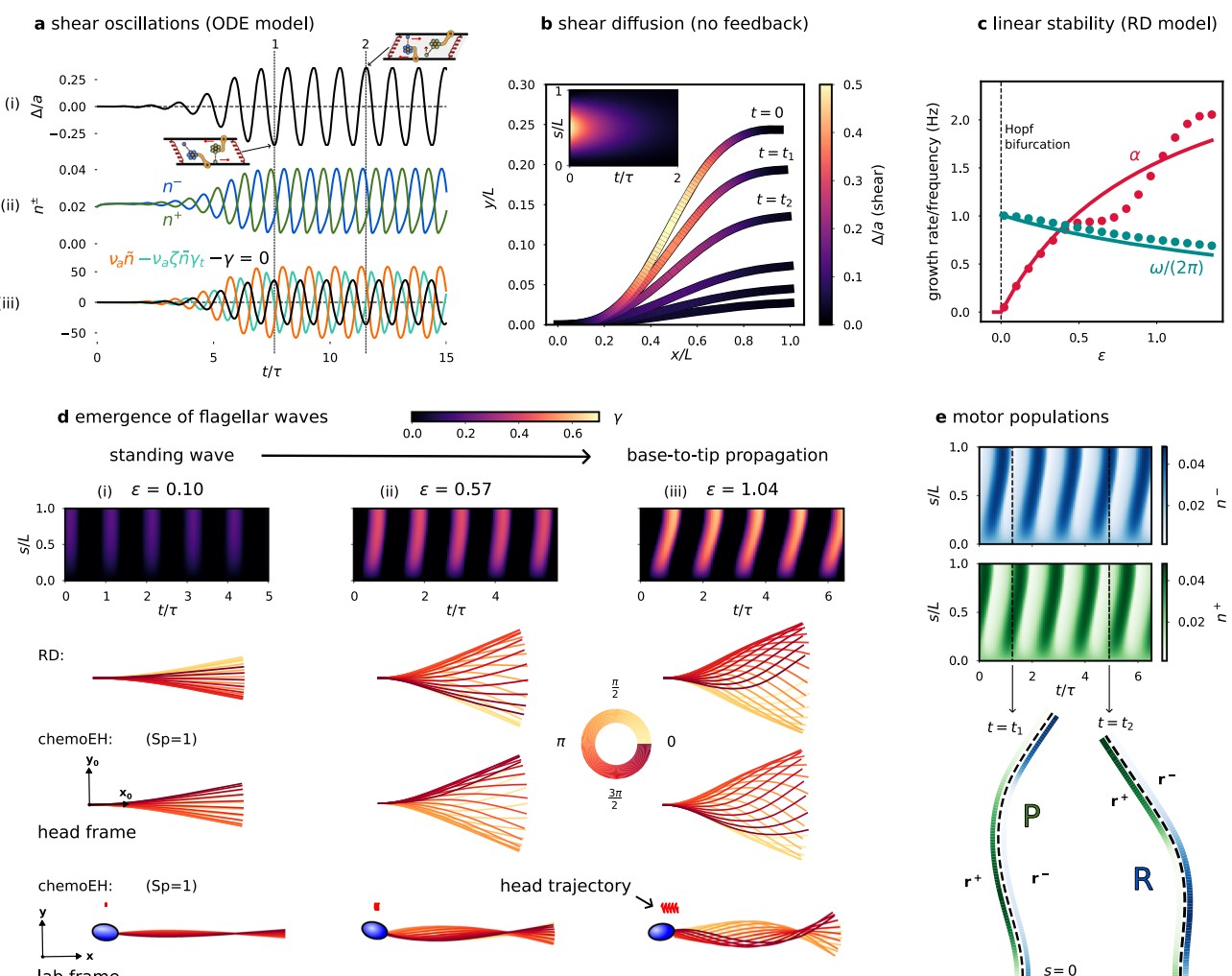

**Fig. 3 | Numerical simulations of RD/chemoEH model. a** Spontaneous oscillations of (i) shear $\gamma = \Delta/a$, (ii) motor populations $n^\pm$ anchored to +/- filament and (iii) forces on an isolated element (see Fig. 2b). **b** Diffusion of shear from an initially curved configuration $\gamma(s) = [1 - \cos(2\pi s/L)]/4$ according to $\gamma_t = D\gamma_{ss} - E\gamma$ in the absence of sliding-controlled feedback. **c** Predictions (solid lines) of growth rate $\alpha$ and frequency $\omega$ of oscillations of the RD model from linear stability theory compared with simulations (filled circles) as a function of the bifurcation parameter

$\epsilon = (\mu_a^{crit} - \mu_a)/\mu_a^{crit}$. Other parameter values $(\mu, \eta, \zeta) = (100, 0.14, 0.3)$. **d** Simulations of the RD model for increasing activity $\epsilon$, showing progression from (i) standing waves for small $\epsilon$ to (iii) base-to-tip progression for $\epsilon = O(1)$. Simulations of the chemoEH model (Sp = 1) at the same parameter values in both the head frame and lab frame. **e** Oscillations of the motor populations $n^\pm$ for the case **d**(iii). Time $t_1$ shows a principal (P) bend and time $t_2$ a reverse (R) bend.

length of the flagellum to the hydrodynamic length scale $l^h = (B\tau/\xi_n)^{\frac{1}{4}}$ over which external viscous forces become comparable with the force required to bend the flagellum in the absence of shear resistance; $\xi_n$ is the resistive-force drag coefficient for motion normal to the flagellum[42], see Section S1 in SI. In these simulations, we have set Sp = 1, using the same parameters otherwise, to highlight the similarities with the RD system. In the head frame $(\mathbf{x}_0, \mathbf{y}_0)$ in Fig. 3d, the beating patterns of RD/chemoEH are indistinguishable (up to an arbitrary phase), and characterised by a base-to-tip wave propagation. We shall explore this further in a later section. Viewed from the lab-frame the time-reversible standing wave in Fig. 3d(i) does not lead to swimming motion (bottom row), whereas in Fig. 3d(iii) a net motion is observed in the bottom row, in the opposite direction to wave propagation, as expected from low Reynold's number hydrodynamics[9,10]. Finally, the motor populations $n^\pm$ exhibit out-of-phase propagating waves of attachment/detachment, shown in Fig. 3e (simulated with parameters corresponding to case (iii) of Fig. 3d). A principal/reverse bend is shown at the times marked $t = t_1/t = t_2$ driven by the green/blue motors on the plus/minus filament, using the same motor colour scheme as in Fig. 2.

## The sliding-controlled flagellar reaction-diffusion system fits animated patterns of eukaryotic flagella

We compared nonlinear simulations of the RD model with experimental data available from previous studies[17,47] in Fig. 4. Specifically, we studied the flagellar beating patterns of a swimming bull spermatozoon[47], and both wild-type and *mbo2*-mutant *Chlamydomonas reinhardtii* cilia that were isolated from the cell body, demembrenated and reactivated[17]. Whilst the flagella (or cilia) of these organisms share the axonemal structure in Fig. 2a, they differ in length ($L \approx 10\,\mu m$ for *C. reinhardtii* vs. $L \approx 50\,\mu m$ for bull sperm), and accessory structures surrounding the axoneme. Moreover, in contrast to the asymmetric wild-type beat, *mbo2*-mutant *C. reinhardtii* flagella are known to beat with an almost symmetric waveform (also at lower frequency and with a shorter flagellum), causing them to swim backwards[17]. Furthermore, it has been shown that the static (asymmetric) and dynamic (oscillatory) components of the *C. reinhardtii* beat are controlled separately[56]. If the static action of motors is represented by a constant curvature $C$ in the governing equations, i.e., $\gamma \rightarrow \gamma - Cs$, then since the time derivative $\gamma_t$ and the second space derivative $\gamma_{ss}$ remain invariant, the dynamics of the RD model are

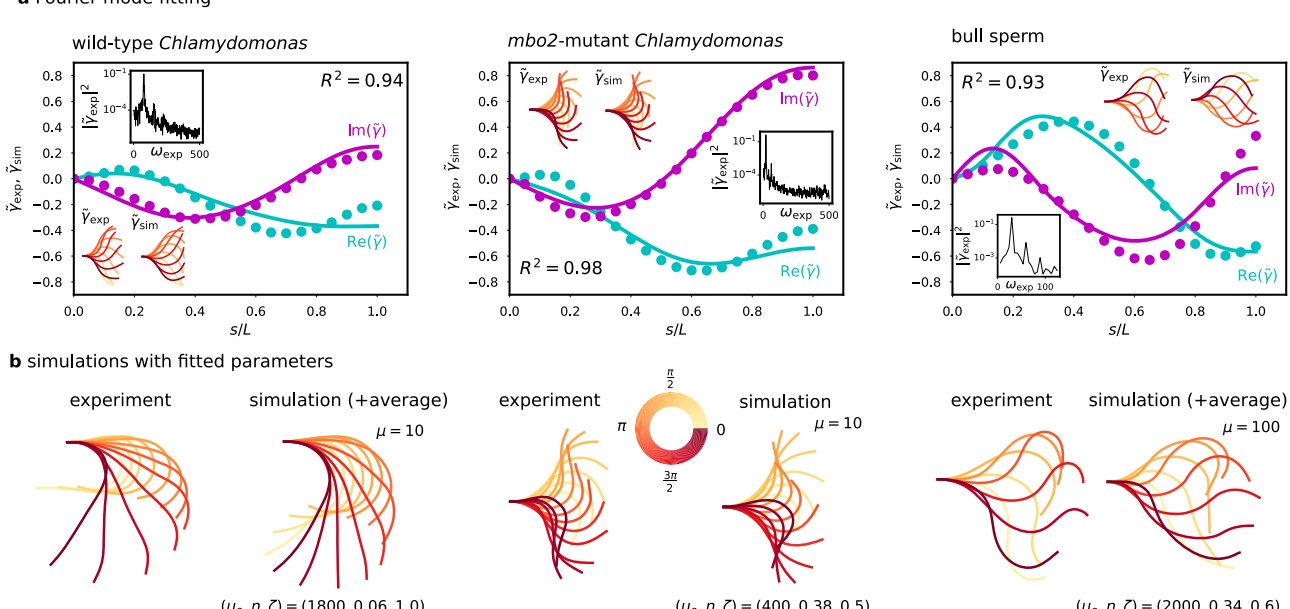

**Fig. 4 | Fourier mode data fitting. a** Comparison of representative experimental ($\tilde{\gamma}_{exp}$, filled circles) and simulated ($\tilde{\gamma}_{sim}$, solid lines) Fourier modes of wild-type/*mbo2*-mutant *Chlamydomonas reinhardtii* and bull sperm, real and imaginary parts. Inset are the reconstructed fundamental mode shapes and strongly peaked experimental power spectra. **b** Comparison of experimental and simulated beating patterns

corresponding to fitted modes in **a**, with fitting parameters ($\mu_a, \eta, \zeta$). The experimental time average shear $\bar{\gamma}_{exp}$ has been added to the wild-type *C. reinhardtii* and bull sperm waveforms for visualisation, so that $\bar{\gamma}_{exp} + \tilde{\gamma}_{exp/sim} \exp(i\omega_{exp/sim}t) + c.c$ is shown over one period of oscillation. This is not done for the *mbo2*-mutant, however, since the beat is approximately symmetric, i.e., $\bar{\gamma}_{exp} \approx 0$.

superposed on the static part of the beat, consistent with being separately controlled[56]. Hence, for simplicity, we focus here on fitting only the symmetric oscillatory part of the beat.

Simulations were carried out varying the three parameters ($\mu_a, \eta, \zeta$), where $\eta = \pi_0 \tau$, which characterise the properties of the dynein molecular motors (see Methods). We used order of magnitude estimates for $\mu$, which combines passive elastic properties of the flagellum that have been estimated previously. For bull sperm, $\mu \approx 100$ has been calculated[26,27]. For *C. reinhardtii*, ref. [57] gives $a^2K \approx 80$pN/rad and $B \approx 840$pN μm, from which we estimate $\mu \approx 10$. The difference in $\mu$ can mostly be attributed to the fact that $\mu \sim L^2$[49].

We compared experimental waveforms with simulations through their Fourier transform in time (see Methods). Given the experimental fundamental Fourier mode $\tilde{\gamma}_{exp}(s)$ at the dominant frequency of oscillation $\omega_{exp}$, and the corresponding mode of a simulation $\tilde{\gamma}_{sim}(s)$, we quantify the quality of fit via

$$R^2(\tilde{\gamma}_{exp}, \tilde{\gamma}_{sim}) = 1 - \frac{\sum_{i=1}^{N} |\tilde{\gamma}_{exp}(s_i) - \tilde{\gamma}_{sim}(s_i)|^2}{\sum_{i=1}^{N} |\tilde{\gamma}_{exp}(s_i)|^2}. \quad (9)$$

where the $s_i$ mark discrete points of arclength along the flagellum, similarly to ref. [16]. We chose the simulation corresponding to the

Fourier mode with the highest $R^2$ score over all parameter values as our fitted result.

The real and imaginary parts of three example simulated modes (solid lines) are compared with the corresponding experimental modes (filled circles) in Fig. 4a, with scores $R^2 = 0.94/0.98/0.93$ for wild-type/*mbo2*/bull sperm. Inset are the reconstructed fundamental modes and experimental power spectra showing a large peak at the fundamental mode (simulated waveforms equally display a large peak at the fundamental mode, see Fig. S3b). Figure 4b directly compares the experimental beating pattern with nonlinear simulations from the RD model. The accuracy of fitting across the three cell types is striking given the large differences of their beating envelope and the reduced number of fitted parameters.

Across $n = 10$ wild-type and *mbo2*-mutant *C. reinhardtii* waveforms[17], we found the fitting score $R^2$ and the estimated values of the three fitted parameters ($\mu_a, \eta, \zeta$) (see Table 1). We calculated $\epsilon = 21.6/19.1/5.92$ using the average fitted parameters for wild-type/*mbo2*/bull sperm, respectively, suggesting the flagellar beat occurs very far from equilibrium in the strongly nonlinear regime. Between wild-type and *mbo2*-mutant *C. reinhardtii*, the estimated parameters show comparable values for $\zeta$, related to the internal drag coefficient of motors, but suggest that *mbo2* mutants have lower activity and a larger duty ratio than their wild-type counterparts (although the relative distance to the bifurcation $\epsilon$ is similar). Videos of the beating cycles at the average fitted parameter values in Table 1 can be found in the Supplementary Information. Supplementary Video 1 shows the simulations of the RD model for the wild-type and *mbo2* mutant *C. reinhardtii* parameters, while Supplementary Video 2 shows the simulated trajectory of bull sperm using the fitted parameters and fixing the sperm number Sp = 1 for nonlinear chemoEH simulations.

**Optimal swimming speeds are observed when viscous moments are small and the flagellar reaction-diffusion system is valid**
In this section, we numerically examine the region of validity of the RD system, and how the relative amount of external and internal dissipation affects swimming performance. Figure 5a depicts simulations of

**Table 1 | Estimated parameter values ($\mu_a, \eta, \zeta$) and best $R^2$ fitting scores (mean ± standard deviation) comparing experimental waveforms[17,47] with simulations of the RD system: $n = 10$ wild-type (wt) *C. reinhardtii*, $n = 10$ mbo2-mutant *C. reinhardtii*, $n = 1$ bull sperm. The value of $\mu = a^2KL^2/B$ was fixed as indicated**

|  | wt ($\mu = 10$) | mbo2 ($\mu = 10$) | bull sperm ($\mu = 100$) |
|---|---|---|---|
| $R^2$ | 0.880 ± 0.044 | 0.967 ± 0.012 | 0.932 |
| $\mu_a = a\rho f_0 L^2/B$ | 1570 ± 377.0 | 490.0 ± 192.1 | 2000 |
| $\eta = \pi_0 \tau$ | 0.096 ± 0.033 | 0.332 ± 0.103 | 0.34 |
| $\zeta = a/(v_0 \tau)$ | 0.96 ± 0.092 | 0.880 ± 0.044 | 0.6 |

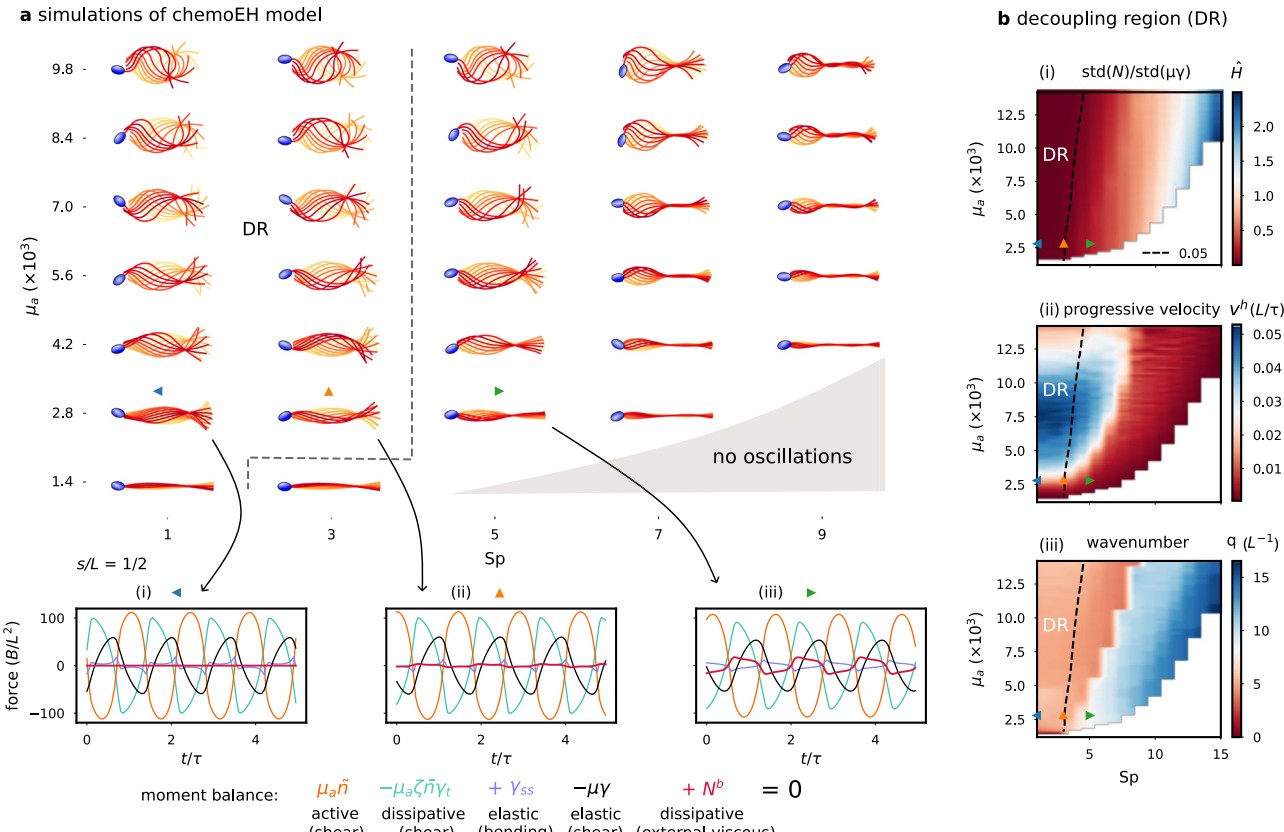

**Fig. 5 | Decoupling of flagellar shaping and swimming. a** Simulations of the chemoEH model in (Sp, $\mu_a$) parameter space. Moment balance at $s = L/2$ (below) showing increase in contribution of $N^b$ with increasing Sp (blue/orange/green triangles). **b** (i) Contribution of viscous moments $N^b$ relative to elastic restoring moments $\mu\gamma$ quantified by the ratio of standard deviations $\hat{H}$. Decoupling region (DR) demarcated by $\hat{H} = 5\%$. (ii) Progressive velocity $v^h$ peaked in the decoupling region. (iii) Wavenumber $q$, divided into left (red) and right (blue) regions of $q \approx 2\pi$ and $q \approx 4\pi$ respectively.

the chemoEH system over a single beating cycle showing base-to-tip wave propagation, for varying sperm number Sp and motor activity parameter $\mu_a$. In these simulations, we have not neglected the contribution of viscous moments $M^v(s)$ to the internal moment balance. We model viscous forces using local resistive-force theory for slender bodies[10,42]. The viscous moment is related to the internal contact force $N(s)$ in the normal direction to the flagellum through $M_s^v = N$. Moreover, it is related to the rate of external viscous dissipation through $P^v = \int_0^L N\gamma_t \, ds$ (see Section S2 in SI). The balance of moments for the chemoEH model, Eq. (3), now reads

$$\mu_a \zeta \bar{n} \gamma_t = \gamma_{ss} - \mu\gamma + \mu_a \tilde{n} + N^b, \quad (10)$$

expressed here in non-dimensional quantities where $N^b = NL^2/B$ and we have made the changes $s \to s/L$ and $t \to t/\tau$. This equation is supplemented by the force balance conditions on the flagellum, as used in previous works[26,27,53], and cell body boundary conditions at $s = 0$ (see Section S1 for details). The balance of moments, Eq. (10), is shown at the point $s = L/2$ over three periods of oscillation in Fig. 5a(i)–(iii) for Sp = 1/3/5 (blue/orange/green triangles) with fixed $\mu_a = 2800$ and $\mu = 100$ (bottom row). For Sp = 1, the contribution of viscous moments (through $N^b$ in red), and hence external dissipation, is visually negligible compared to the other terms (Fig. 5a(i, ii)), consistent with our assumption in deriving the RD model. As Sp increases the contribution of $N^b$ becomes more significant (Fig. 5a(iii)).

To quantify the hydrodynamic contribution in shaping the beat, we introduce the ratio $\hat{H} = \text{std}(N^b)/\text{std}(\mu\gamma)$ of the standard deviations (over many periods and over arclength) of the contact force $N^b$ and the restoring force $\mu\gamma$ (see Fig. 5b(i)). The dashed line marks the *decoupling*

*region* (DR), taken to be where $\hat{H}$ is less than 5%, approximately passing through the orange triangle simulation at Sp = 3 (Fig. 5a(ii)), so that the blue/green triangles are representatives of inside/outside the decoupling region, corresponding to Fig. 5a(i) and (iii), respectively. As activity $\mu_a$ increases, the oscillations increase in amplitude, and therefore the resistance to shear is also higher in the denominator of $\hat{H}$ (Fig. 5b(i)). This allows for a wider range of sperm numbers in the decoupling region and results in the approximately straight line relationship visible in Fig. 5b(i). For such small relative contributions, the effect of viscous moments can be neglected in comparison with moments from elastic shear resistance, and the waveform is well approximated by the RD equations only. We have seen examples of this likeness previously in Fig. 3d and further evidence can be found in Fig. S3a showing the result $R^2(\tilde{\gamma}_{\text{sim}}^{\text{CEH}}, \tilde{\gamma}_{\text{sim}}^{\text{RD}})$ of applying the above fitting procedure to compare the simulated chemoEH and RD waveforms, finding $R^2 \approx 1$ in the decoupling region, justifying our choice of 5% for the demarcation line.

We characterise the swimming ability by the head velocity $v^h$ in the swimming direction (see Fig. 5b(ii)). The highest swimming performance is found at low Sp, and is non-monotonic in $\mu_a$ (as reported in ref. 58). Crucially the peak values lie within the decoupling region, and these waveforms qualitatively resemble swimming spermatozoa in low viscosity[47,59]. For larger Sp, swimming velocity is reduced, with the waveforms characterised by large angular deviations of the head and little forward motion. Swimming performance is correlated with the wavenumber $q$, which divides the parameter space into left (red) and right (blue) regions in Fig. 5b(iii), with wavenumbers $\approx 2\pi$ to the left (corresponding to a wavelength of $L$), and $\approx 4\pi$ to the right (with wavelength $L/2$). Crossing from red to blue in Fig. 5b(iii) leads to a

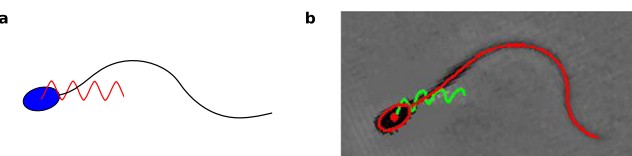

**Fig. 6 | Fitted motor parameters reproduce bull sperm swimming motion.**
Qualitatively similar distance travelled per beat, amplitude, frequency, and shape of the head trajectories of (**a**) simulated bull sperm from the full flagellar chemoEH system at the fitted motor parameters ($\mu_a, \eta, \zeta$) derived from the simplified RD system, as given in Table 1, and Sp = 1 (not fitted), and (**b**) image captured from the Supplementary Video 1 in ref. [47] (adapted with permission from ref. [47], Copyright 2020 Royal Society), from which the bull sperm data used for fitting was extracted. Note that the empirical average $\bar{\gamma}_{exp}$ of the beat, leading to the curved trajectory, was not included in (**a**) to highlight the true contribution of the emergent symmetric (oscillatory) waveform mode on the swimming motion, as predicted from the full chemoEH system.

catastrophic reduction of swimming speed in Fig. 5b(ii). All waves propagated from base-to-tip in the simulated parameter space for the chemoEH system, in further agreement with the RD model.

### Fitted motor parameters from the flagellar reaction-diffusion system reproduce bull sperm swimming kinematics

Here, we provide a direct comparison beyond the emergence of the flagellar beat in Fig. 4, at the level of the swimmer's self-organised trajectory. For this, we simulate the full flagellar chemoEH system at the fitted motor parameter values for bull sperm in Table 1, as derived from the simplified flagellar RD system above, setting the sperm number low (Sp = 1), rather than fitting it, to locate the system within the decoupling region ($\bar{H} = 3.4 \times 10^{-4}$, see Fig. 5b(i)). Figure 6 shows the qualitative similarity of the predicted bull sperm swimming trajectory with experiments[47]. An excellent agreement is observed for the distance travelled per beat, shape of the head trajectory, amplitude and frequency of head yawing oscillations resulting from the self-organised flagellar beating. Unlike Fig. 4b, the static asymmetric part of the beat $\bar{\gamma}_{exp}$ has not been added in Fig. 6 to highlight the contribution of the emergent symmetric (oscillatory) part of the simulated beat. As a result, a straight trajectory is observed in Fig. 6 rather than the slightly curved trajectory of the experiment due to the presence of an asymmetric static component of the waveform (Fig. 4b)[43].

## Discussion

### Flagellar patterning can be explained by a minimal reaction-diffusion system invoking the balance of four moments

We have seen that a balance of the motor force, motor friction and elastic shear resistance is sufficient to produce sustained shear oscillations in a small flagellar element. Oscillations of a flagellar section then exert moments on neighbouring elements that are resisted by bending elasticity. This is the flagellar reaction-diffusion (RD) model for animated patterns, which displays a Hopf bifurcation to standing wave oscillations at a critical value of the activity parameter $\mu_a$. As the bifurcation parameter $\epsilon$ increases from zero, the oscillations increase in amplitude and begin to propagate from base-to-tip with increasing wavespeed, providing a plausible mechanism for generating self-organised metachronal shear oscillations from oscillatory local elements[14,40], as first hypothesised by Brokaw[13]. Furthermore, by varying the only three molecular motor parameters ($\mu_a, \eta, \zeta$), the RD model produces diverse beating patterns that mimic those of eukaryotic flagella. This was demonstrated by fitting numerically simulated shapes to experimental data from the literature[17,47].

As well as external hydrodynamic forces and the cell body coupling condition, many other effects have been assumed to be negligible; to name only a few: the three-dimensional nature of the beat of bull sperm[60], bending friction in the microtubules[44,61], internal fluid

viscosity[23], basal elasticity[16,62], variable bending stiffness from the flagellar ultrastructure[63], and nonlinearity of the force-velocity relationship of axonemal dynein, which is known to exist for kinesin[64]. We have also not considered any asymmetry of the beating patterns in this work[17], utilising the fact that the static and dynamic components of the beat of *C. reinhardtii* appear to be independently controlled[56] to concentrate only on the oscillatory motion. We have shown, however, that the minimal reaction-diffusion model is sufficient to capture characteristic waveforms within a small range of parameters.

Including other sources of internal dissipation in the RD model would change the ratio of external to internal dissipation and it may be possible to extend the region of validity of the RD model to higher Sp, or equivalently, higher external fluid viscosity. Additionally, it is possible that a higher quality of fit may be achieved if $\mu$ was allowed to vary, but we have restricted our focus here to the unknown motor parameters. The lower average $R^2$ for wild-type vs. *mbo2* (0.88 vs. 0.97) suggests important differences in mechanical properties of the two *C. reinhardtii* cell types, also not accounted for here; it was suggested by the fitting results of ref. 17 that the *mbo2*-mutant should have a 20-fold smaller basal stiffness than the wild-type, for example. Interestingly, although the fitted parameter values $\mu_a$ and $\eta$ for the two *C. reinhardtii* genotypes appear quite different, the average relative distance from the bifurcation $\epsilon = 21.6/19.1$ is similar, suggesting similar beating amplitudes (the supercritical amplitude for a Hopf bifurcation tends to be $\sim \sqrt{\epsilon}$). The lower value of $\mu_a$ for the *mbo2*-mutant could arise from, for example, the inactivity or absence of some of the inner/outer arm dyneins in the axoneme of this mutant.

### Fitting nonlinearity unifies dynein regulatory mechanism in two eukaryotic species

Previous studies fitting model beating patterns to flagellar centerline measurements[16–18] have been based on the fact that near the Hopf bifurcation, the elastohydrodynamic system is well-modelled by its linear terms[20], reducing to the hyperdiffusive equation, $\xi^n \theta_t = -B\theta_{ssss} + af_{ss}^t$, in the absence of molecular motor reaction kinetics. At this level, the feedback mechanism reduces to a linear relation between the motor force and the particular mode of deformation that regulates motor activity (sliding, curvature or normal forces), but which does not contain the *specific details of the cross-bridge reaction kinetics*− the latter is only retained at the nonlinear level[26]. Canonically, $\tilde{f}^m = \chi \tilde{\Delta}$ linearly relates the fundamental mode of the active force $\tilde{f}^m$ to the sliding $\tilde{\Delta}$, where $\chi$ is a (complex) linear response coefficient (see Table S2 in SI). Nonlinear solutions allow, instead, a direct comparison between model prediction (simulations) and experiments, as depicted in Fig. 4b, since the waveform (amplitude, frequency and phase) arises dynamically via the nonlinear saturation of unstable modes. Such direct comparison is not possible with linear models, as the amplitude of beating is undetermined, and thus, generally, fitted for comparison purposes[16–18].

In a seminal work, ref. 16 found for the first time that the beat of free swimming bull sperm were accurately described using the sliding response coefficient $\chi$ (average $R^2 = 0.96$). For the single cell studied here, we find the value of $R^2 = 0.93$, in apparent agreement with this earlier result. However, there are crucial distinctions, in addition to the points discussed above: (i) basal sliding, the subject of many experimental studies[62,65], is not considered here, though deemed essential to find satisfactory fits in ref. 16, where precise values of basal resistance appear to be necessary (appearing as eigenvalues of the linear problem), (ii) our fit uses large-amplitude, highly nonlinear solutions, far from the onset of the Hopf bifurcation, with $\epsilon = 5.92$, in contrast with ref. 16 that focused on the linear regime with oscillations near equilibrium−the linear solutions for the RD model in Fig. 2d are standing waves, highlighting the qualitative differences between the RD model and previous work, (iii) The formulation in ref. 16 requires seven unspecified parameters for the free-swimming case, compared

to only three parameters required by the RD model: the amplitude of the beating (not determined at the linear level, but emergent here), the complex response coefficient $\chi$ and basal response $\bar{\chi}$ (2 parameters each), and the translational and angular drag coefficients of the sperm head, and (iv) the fitted parameters here were not optimised continuously over the parameter space, as is more easily achieved with analytical solutions[16]. Instead, the best fitting shape, with highest $R^2$ score, was found from a set of 5460 simulations. Hence, the space of models we are fitting from here is vastly more constrained (only 3 parameters), no parameters require fine-tuning, as in the case of the basal response coefficient, and higher values of $R^2$ may be possible still for parameter values outside our test set. This result should be viewed in this light; a considerable reduction in model freedom with no significant loss of fitting accuracy.

We have assumed the bull sperm cell is in the decoupling region (Fig. 5b(i)) in order to fit with the RD model. Supposing the decoupling assumption is violated, so that hydrodynamic effects on the emergence of the beat are significant for this cell, it could be the case that these effects (including the effect of the cell body) may be compensated for within our model by tuning the fit parameters. If this was the case, however, then when calculating the overall translation and rotation of the sperm using RFT, see Fig. 6, we would expect these unaccounted for hydrodynamic effects to cause significant deviations from the experimental trajectory, and this is not observed. The value of $\dot{H} = 3.4 \times 10^{-4}$ confirms the chemoEH simulation is in the decoupling region; this is strong evidence that the decoupling assumption applies for this cell.

At the swimming trajectory level, even small errors in each beating cycle can be accumulated into large discrepancies of the swimming trajectory after many cycles. Swimming trajectories that were obtained via the boundary element method[66], a highly accurate hydrodynamic theory, demonstrate how these discrepancies accumulate in the distance travelled per beat even when two or three principal components of the experimentally observed beating pattern, that very accurately capture the waveform, are used; this is in contrast to the chemoEH simulations in Fig. 6. Therefore, it is remarkable that this close match in swimming characteristics has emerged at the microscale just by setting the nanoscale parameters ($\mu_a, \eta, \zeta$) to their fitted values in Table 1, and is derived from a simplified flagellar RD system that does not account for hydrodynamic interactions. Further experimental tests with larger numbers of cells are required, however, to determine if this result applies more generally than in the case of this particular cell; similarly for the results of ref. 66. In all, the framework developed here allows for the first time direct hypothesis testing beyond the emergence of the flagellar beat (Fig. 4), encompassing the resulting self-organised swimming motion observed in the laboratory fixed frame of reference (Fig. 6), to gain better understanding of the validity of a given model, thus closing the modelling cycle.

Wild-type and *mbo2*-mutant *C. reinhardtii* were previously fitted very well by a dynamic curvature response coefficient, i.e., proportional to $\theta_{st}$, averaging $R^2 = 0.95/0.95$ for wild-type/*mbo2*-mutant by ref. 17. Recently, this method was commendably used on a much larger waveform data set for *C. reinhardtii* axonemes under various experimental and genetic perturbations, and similarly excellent fits were reported[18]. The *sliding control mechanism*, however, was not able to produce good fits within the framework presented in ref. 17, only averaging $R^2 = 0.49/0.72$ for wild-type/*mbo2*-mutant see (Fig. 7). Crucially, since the linear fitting method uses a generic sliding-controlled response coefficient[16,17,20], the poor fitting observed for sliding control in ref. 17 would seem to be evidence against any sliding-controlled mechanism—indicating instead that different motor control mechanisms by which the beat is generated may exist in the 50 μm long flagella of bull sperm compared to the 10 μm cilia of *C. reinhardtii*[17]. A key result presented here, however, is a dramatic improvement in $R^2$ values for a sliding-controlled mechanism, with the average $R^2 = 0.88/0.97$ for

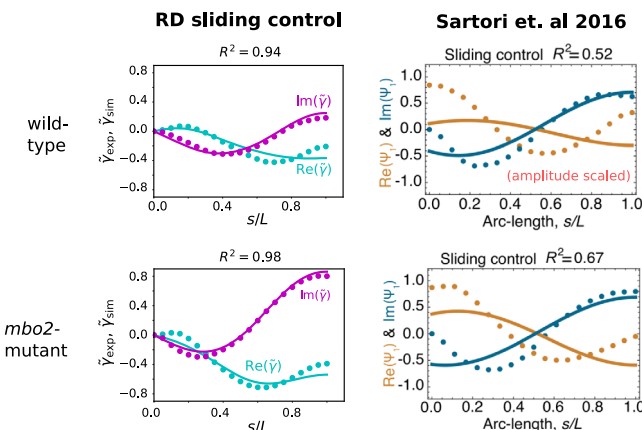

**Fig. 7 | Comparison between the fitting accuracy of the geometrically exact RD sliding-controlled model (left column) and linearised elastohydrodynamic sliding-control model[17] (right column).** Real/imaginary parts of the fundamental Fourier mode of typical theoretical (solid lines) and experimental (dotted lines) beats of wild-type and *mbo2*-mutant *C. reinhardtii* fitted with: (i) the RD model with tug-of-war sliding-controlled motor kinetics and geometrical nonlinearities retained (left column), and (ii) a complex sliding-controlled linear response coefficient $\chi$ and free amplitude, adapted from ref. 17 (CC BY 4.0, right column).

the same waveform data set used in ref. 17. The improvement is seen to be striking in the direct comparison with the RD model in Fig. 7. A summary of the distinctions discussed in this section between the fitting procedure presented here and previous fittings[16,17] is provided in Table S2 in the Supplementary Information.

Although it is known that weakly nonlinear waveforms should resemble the linear modes used in the previous fitting studies[67], our results highlight that there can be large differences between the shape of patterns generated in the linear regime close to the bifurcation, and the strongly nonlinear regime where our fitted average values lie ($\epsilon = 21.6/19.1$ for wild-type/*mbo2*). We thus argue that it is essential to include molecular motor and geometric nonlinearities when collecting evidence for particular flagellar control mechanisms. Seeking simplifications such as the high internal dissipation limit can reduce complexity and enable this. We have shown that nonlinear solutions of a sliding-controlled RD model with a specific motor cross-bridge reaction-kinetics can in fact capture accurately the beating patterns of *C. reinhardtii*, unifying two distant eukaryotic species (bull sperm and *C. reinhardtii*) under a single motor-control mechanism. Furthermore, as molecular mechanisms for curvature-sensing by dynein are at present unknown (the small strains involved make this hard to envisage, as recognised in refs. 17,18), whereas the load-dependent detachment of dynein suggests a physical basis for sliding-control, we argue that the sliding-control hypothesis, of the two, should be favoured as the more parsimonious given current evidence. Regulation of dynein by transverse forces, a way by which dynein may sense curvature indirectly[28,30], was described as unlikely in *C. Reinhardtii*[17] despite good fitting scores, due to large intraspecies variation in the fitted parameter values. Further research is needed to confirm or refute this finding for large amplitude nonlinear beating, in light of the current work. A dynamic flutter instability[32,33] also offers an attractive and simple alternative mechanism, but it remains to be seen whether this model can account quantitatively for experimental beating patterns, including the observed high internal dissipation by the flagellum.

We may further speculate about the possibility of a universal, minimal pattern formation mechanism for animating cilia and flagella that is capable of robustly generating travelling waves, independently of, and without relying on, the external fluid viscosity, when small (see below). The intrinsic internal dissipation may equip cilia and flagella of species living in low viscosity environments, such as green algae,

aquatic microorganisms, and external fertilizers[47,68] with the capacity to generate autonomous, hydrodynamic propulsion—not possible with the external fluid dissipation alone (Fig. 3d(i)). The quality of fitting suggests that the experimental beats in low viscosity considered here may indeed be operating in the hydrodynamic decoupling region, uninfluenced by external viscous friction, consistent with previous energy calculations[18,44,45].

## Internal dissipation enables progressive swimming in low viscosity environments

Figure 5 suggests a possible explanation for the seeming inefficiency of the high internal dissipation regime. For a fixed, low activity level $\mu_a$, increasing Sp increases the external hydrodynamic dissipation, but the work done on the fluid is not propulsive, and therefore does not induce any increase in swimming speed $v^h$. At low sperm number much of the energy converted from chemical energy (ATP) to mechanical work is dissipated internally, however, the result is a faster swimmer (Fig. 5b(ii)), with a more hydrodynamically efficient stroke. This is due to the symmetry-breaking instigated by the intrinsic internal dissipation, which transforms time-reversible standing waves for low activity (Figs. 3d(i) and 5), and zero net propulsion, into propulsive travelling waves for increased activity (Figs. 3d(iii) and 5). It is noteworthy that the difference in the mathematical structure of the governing equations to previous works employing elastohydrodynamics of flagella leads to very different solutions and instability types, namely, standing waves which do not lead to swimming (see Fig. 3d(i)). Compare this with Camalet and Julicher [22], where propagation is still observed for small amplitudes and the direction depends on the boundary conditions. Instead, symmetry breaking in the RD model is only observed for increased activity far away from the instability (Fig. 3d(iii)).

The internal dissipation thus allows the flagellum to break Purcell's scallop theorem, and its time-reversibility constraint, before the hydrodynamic dissipation becomes relevant, when Sp is low. As discussed above, this may be a critical mechanism allowing the emergence of autonomous travelling waves in low viscosity, as non-zero hydrodynamic propulsion in elastohydrodynamic systems operating near-zero sperm number are, generally, not possible[37,50,55]. Passive filaments and filament-bundles (without internal dissipation), actuated externally, require Sp ≈ 2 for elastohydrodynamic dissipation to symmetry-break effectively stiff filaments (non-propulsive, standing-waves) into flexive travelling waves with net propulsion. In this case, the propulsive force reaches a maximum, before decaying due to the excess in viscous friction as Sp continuously increases[37,50,55]. Similarly, in the limit of high internal dissipation, the hydrodynamic propulsion reaches a maximum for increasing activity (see Fig. 5b(ii) for Sp = 1 and increasing $\mu_a$). As such, the motor activity for the RD model can play a similar role as the sperm number for propulsive elastohydrodynamics[37].

Although we have used local resistive-force theory here, the distribution of swimming speeds shown in Fig. 5b does not change appreciably with the inclusion of non-local hydrodynamics through slender-body theory, as was reported recently in ref. 58. The wavelength of beating patterns in the decoupling region is approximately equal to the length of the flagellum $L$ and this is observed in *C. reinhardtii* (see Fig. 3c of ref. 18) and bull sperm[43]. These patterns have the benefit of being robust to external hydrodynamic perturbations, and to a significant loss of swimming velocity when crossing from left to right in Fig. 5b(iii), although this abrupt change could be due to the specific exponential form of the detachment kinetics used here, Eq. (2). At higher Sp, the wavelengths are shorter ($\approx L/2$). This observation is consistent with experiments with human spermatozoa in high viscosity[59]; recall that $\mathrm{Sp} \sim (\xi^n)^{\frac{1}{4}}$.

Figure 5a suggests, however, that the wave amplitude decreases as the wave propagates for increasing Sp, which is not observed experimentally[11]. Moreover, increasing the viscosity experimentally

does not always lead to a drop in swimming speed[11], as Fig. 5b(ii) would suggest. Hence our chemoEH model is not capturing some aspects of the beating for higher Sp, and further research is required to investigate flagellar waveform modulation by viscosity[59]. For example, there may be an effect of curvature, in addition to sliding on the motor recruitment, such as used in ref. 36, where higher wavenumber beating is possible without reduced amplitude of propagation. A combined sliding/curvature control approach was also considered in ref. 27 for asymmetric waveforms, whereas the effective curvature-controlled moments model-type[24] gave promising results recently for human sperm in high viscosity, consistent with this idea[69]. The limited applicability of the chemoEH model to the high viscosity regime, however, is less relevant to the results of this paper, which focus on the hydrodynamic decoupling region and the RD model.

## Outlook

By isolating the essential elements of the flagellar beat in a minimal model, we discovered that reaction-diffusion dynamics account well for the observed flagellar beating patterns. The oscillatory dynamics are analogous to those observed in chemical systems like the BZ reaction—oscillations can persist in an isolated, small section of the system[2]. With diffusive coupling between subsystems, metachronal waves can emerge, inducing animated beating patterns in the case of a flagellum, analogous to target waves in the BZ reaction. Since hydrodynamics applies only a small perturbation to the beating pattern in the decoupling region, there is the potential in future work to apply phase-reduction techniques to study the hydrodynamic synchronisation of many self-organising flagella[70–72], driven by reaction-diffusion dynamics.

The simplified RD model may be of interest to many researchers studying molecular motor organisation, or in mathematical biology more generally, since analytical progress and numerical simulations become easier in this framework, avoiding complexities arising from the hydrodynamic coupling. The RD framework can thus be used as a fundamental building block for future hypothesis testing, and generalised to the multi-physics of flagellar interactions, to include, for example, transmembrane dynamics of ion channels that modulate asymmetric gaits of the beat via calcium fluxes, stochastic oscillations in cilia and flagella, cell chemotaxis and rheotaxis, flagella rheology, signalling and multi-flagellar interactions[10,11,14], to name a few. The fact that the molecular shaping of the emergent flagellar beat is decoupled from the hydrodynamics of swimming (in the decoupling region) allows researchers to exploit a diversity of low Reynolds number hydrodynamic methods to solve for the kinematics of flagellated swimming separately[10] (without requiring full chemoEH simulations in Fig. 6); using, instead of ad-hoc prescribed waveforms that are typically invoked[42], the self-organised animated beating pattern (Supplementary Video 2) obtained from the RD tug-of-war model that has been validated against experimental observations, and as such, it may appeal to the micro-hydrodynamics community in general.

Our reaction-diffusion framework may not be limited to flagellar beating. For example, oscillations have been observed in-vitro for microtubules[73] and actin filament-bundles[36] in the presence of free kinesin and myosin motors, respectively. In ref. 36, the authors employed a phenomenological model that also neglected external viscous friction over internal friction, and compares well with the observed oscillations of actin bundles. They suggest that the source of this internal friction could be the transiently cross-linked myosin motors (modelled by a shear friction coefficient $\xi^a$), in further congruence with our results in which the motor friction was explicitly modelled. In contrast to the tug-of-war reaction kinetics used here, the myosin binding/unbinding kinetics were based on a curvature controlled attachment rate, in addition to advection of myosin molecules along filaments [*ibid*], whereas our dyneins are anchored in place. These elements can be incorporated into the reaction-diffusion

framework described here through simple alterations to the governing equation, Eq. (6). Other possible extensions could allow for easier comparison of molecular motor control mechanisms such as the geometric clutch[28,30] and flutter instability[32,33], or support investigations into the internal rheology of flagella. For this, we include a simple interactive Python implementation of the RD model so that researchers may explore these possibilities further.

As experiments at the nano-scale can be costly and challenging, an alternative route to understanding is the practical design and engineering of artificial swimmers[74], animate materials or Turing patterns more generally[75,76]. We should heed from these studies, however, that distinct mechanisms may bring about the same patterns, and this complicates judging whether models of flagellar beating are truly describing the underlying physics, for which more research will be required. That said, we hope this paper is a step towards an understandable, minimal model of flagellar patterning that is amenable to mathematical analysis, in the spirit of Turing's work[1], and may appeal to researchers interested in pattern forming structures of reaction-diffusion systems at large.

## Methods

### Interpolation of data and Fourier analysis

We obtained the raw data for *Chlamydomonas reinhardtii* (10 wild-type and 10 *mbo2*-mutant trajectories) from ref. [17]. This consists of the tangent angle $\theta_{exp}(s_i, t_j)$ in the lab frame at 19 points of arclength $s_i$ at times $t_j$ with $\Delta t = 1$ms. We then calculated the relative angle $\gamma_{exp}(s_i) = \theta_{exp}(s_i) - \theta_{exp}(s_0)$. We interpolated the signal (using the Python function `interp1d` with cubic splines) at $m = 101$ points of arclength in order to compare with our simulated waveforms. For the bull sperm trajectory (obtained from ref. [47]) the curvature was supplied at 30 points of arclength at time intervals of $\Delta t = 4$ms, so the relative angle $\gamma_{exp}(s_i, t_j)$ was obtained via a single numerical integration before interpolating.

To obtain the fundamental Fourier mode $\tilde{\gamma}_{exp}(s_i)$ of the interpolated waveforms, we Fourier transformed > 20 periods of the signal in time. We obtained the fundamental frequency by finding the peak of the spatially averaged power spectrum, as in refs. [16,17]. The mode shape $\tilde{\gamma}_{exp}(s_i)$ is then extracted in correspondence with this fundamental frequency. Applying the same Fourier decomposition to simulated beating patterns, we then compared the experimental and simulated fundamental Fourier modes with the $R^2$ measure (Eq. (9)). Before calculating $R^2$ we first multiply $\tilde{\gamma}_{sim}$ by a phase factor $e^{i\phi}$ to bring the phases of the simulated and experimental oscillations into alignment.

### Calculating the wavenumber $q$

The wavenumber $q$ of a signal $\theta(s_i, t_j)$ was calculated using the auto-correlation function $A_k = \sum_i \theta(s_{i+k}) \theta(s_k)$, averaged over > 20 time periods. The peaks/valleys of the signal are located where the derivative changes sign; we took the peak of maximal magnitude for the wavenumber (or twice the value of the valley of maximal magnitude if the wavenumber is smaller than one full oscillation).

### Numerical methods

To discretize the continuous system of partial differential equations for computation, we start by sampling $m = 101$ equally spaced points of arclength $\gamma_0 - \gamma_{m-1}$ along the flagellum. For spatial derivatives, we use second-order finite differences, centred on the interior of the domain and sided at the boundaries. The RD system then reads in non-dimensional form

$$\gamma_{t,0} + \gamma_0 = 0 \tag{11}$$

$$\mu\gamma_i - m^2(\gamma_{i+1} - 2\gamma_i + \gamma_{i-1}) + \mu_a(\zeta\bar{n}_i\gamma_{t,i} - \bar{n}_i) = 0 \tag{12}$$

$$\mu\gamma_{m-1} - 2m^2(\gamma_{m-2} - \gamma_{m-1}) + \mu_a(\zeta\bar{n}_{m-1}\gamma_{t,m-1} - \tilde{n}_{m-1}) = 0 \tag{13}$$

$$(n_\pm)_{t,i} - \eta(1 - n_{\pm,i}) + (1 - \eta)n_{\pm,i}e^{f^*(1\mp\zeta\gamma_{t,i})} = 0 \tag{14}$$

where $i$ runs over the interior $m - 2$ points. The $\gamma_0$ term in Eq. (11) is a penalization term that dampens any small deviations from zero at the point $s = 0$. This is a system of $3m - 2$ equations in $3m - 2$ unknowns. It is possible to solve for the time derivatives in the above equations, and step forward in time using the method of lines (MOL), i.e., to use an ODE time-stepping algorithm on the $3m - 4$ unknowns. An interactive Python implementation in a Jupyter notebook can be found at the github link provided. We found simulations to be more numerically stable at high $\mu_a$ with a DAE (differential-algebraic equation) solver. These are used for systems that contain a combination of differential equations and algebraic equations. This method also generalises straightforwardly to the chemoEH model (see Section S1.6 in SI), where we have constraint equations that are algebraic.

We used the IDA solver in the Sundials suite[77] to solve the DAE system. The left-hand sides of Eqs. (11)–(14) are supplied as the residuals to be minimised by the solver. Although written in the C language, interfaces to the solver are available through higher level languages like Python and Julia. All simulations were started from a small Gaussian perturbation centred at the midpoint, $\gamma(s,0) = 0.001 \exp(((s - 0.5)/0.1)^2)$, with the bound motor populations set to the constant equilibrium value $n_\pm(s, 0) = n_0$. We stepped forward each simulation for many periods (100 non-dimensional time units) in order to ensure convergence to the limit cycle. Simulations were carried out using the parameters in Table S1 in the Supplementary Information. The step sizes in parameter space for $(\mu_a, \eta, \zeta)$ were (100, 0.04, 0.1), (1000, 0.04, 0.1) for $\mu = 10, 100$, with a total of 3250 at $\mu = 10$, and 2210 simulations at $\mu = 100$.

## Data availability

All data used in this paper can be found in two repositories made available in previous studies: •https://doi.org/10.5061/dryad.0529j[17]• https://doi.org/10.7910/DVN/CPAPV1[47].

## Code availability

An implementation of the reaction-diffusion model is provided in a Jupyter notebook written in Python. The code is available via the GitHub repository: https://github.com/polymaths-lab/reaction-diffusion-flagella, or on Zenodo[46].

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

## Acknowledgements

We express our gratitude to the authors of refs. 17,47 for making the data from their studies freely available, without which we would not have been able to proceed with this study. HBG acknowledges funding from the Engineering and Physical Sciences Research Council (EPSRC) DTP studentship for JFC's PhD, with Grant Code: EP/R513179/1. The numerical work was carried out using the computational and data storage facilities of the Advanced Computing Research Centre, University of Bristol - http://www.bristol.ac.uk/acrc/.

## Author contributions

J.F.C. conducted the research, H.B.G. supervised the research, J.F.C. and H.B.G. wrote the manuscript.

## Competing interests

The authors declare no competing interests.
