## [Peer Review File · Nature Communications]

REVIEWERS' COMMENTS

Reviewer #1 (Remarks to the Author):

I thank the authors for their authoritative and compelling rebuttal of my comments. I am fully convinced now that this work does indeed provide considerable new insight and is not an extension of existing ideas. The work and the presentation are of a very high standard. The technical ideas are also made accessible so that they may be widely tested. I recommend publication without any further changes.

Reviewer #2 (Remarks to the Author):

The authors have addressed my main concerns. Two minor points:

1. The authors suggest in lines 860-867, there is no plausible mechanism for curvature effect on dynein. I agree that it is unlikely that dyneins sense curvature directly, but Lindemann [30] has pointed out (and Bayly and Wilson [32]) affirm), that tensile or compressive forces in doublets will pull doublets together or push doublets apart in regions of high curvature, to an extent that depends on the directions and magnitudes of the forces and curvatures. It seems plausible that this inter-doublet separation, which is affected by curvature, can affect dynein forces.
2. Finally, following their argument on lines 866-868, it is worth noting that the most parsimonious explanation of all is the flutter instability under steady dynein loading ([34], [35], which requires no feedback at all).

Response to Reviewers

We sincerely thank the reviewers once again for their dedicated and specialist time and contributions.

Reviewer #1

I thank the authors for their authoritative and compelling rebuttal of my comments. I am fully convinced now that this work does indeed provide considerable new insight and is not an extension of existing ideas. The work and the presentation are of a very high standard. The technical ideas are also made accessible so that they may be widely tested. I recommend publication without any further changes.

We are humbled by the referee's report, and sincerely grateful for their reconsideration and full endorsement of our study for publication.

Reviewer #2

The authors have addressed my main concerns. Two minor points:

1. The authors suggest in lines 860-867, there is no plausible mechanism for curvature effect on dynein. I agree that it is unlikely that dyneins sense curvature directly, but Lindemann [30] has pointed out (and Bayly and Wilson [32] affirm), that tensile or compressive forces in doublets will pull doublets together or push doublets apart in regions of high curvature, to an extent that depends on the directions and magnitudes of the forces and curvatures. It seems plausible that this inter-doublet separation, which is affected by curvature, can affect dynein forces.

We thank the reviewer for this important comment. It is true that curvature sensing may come about in this indirect way, and we have added a sentence to that part of the discussion to highlight this. We were implicitly basing our arguments at this point on the findings of Sartori et. al [19], which argued against normal force control for *Chlamydomonas*. We have now made this explicit, as well as highlighting the need for similar research with nonlinear beating using the geometric clutch hypothesis to investigate this important aspect in future, exploiting, for example, the simplifications associated with the proposed reaction-diffusion limit.

2. Finally, following their argument on lines 866-868, it is worth noting that the most parsimonious explanation of all is the flutter instability under steady dynein loading ([34], [35], which requires no feedback at all).

Many thanks for this. We understand this perspective, the flutter instability certainly provides a very simple explanation. However, as far as we know at present, the model has not demonstrated quantitative fits with experimental data in eukaryotic species as we are discussing here, or addressed how the observed internal dissipation may be significant. We have added a sentence raising this point to the same paragraph. We wish to thank the reviewer once again for all comments, they have truly improved the scope of our analysis.